# Physiological Responses of Post-Dietary Effects: Lessons from Pre-Clinical and Clinical Studies

**DOI:** 10.3390/metabo11020062

**Published:** 2021-01-20

**Authors:** Christy Yeung, Irisa Qianwen Shi, Hoon-Ki Sung

**Affiliations:** 1Translational Medicine Program, The Hospital for Sick Children, Toronto, ON M5G 0A4, Canada; christy.yeung@mail.utoronto.ca (C.Y.); irisa.shi@mail.utoronto.ca (I.Q.S.); 2Department of Laboratory Medicine and Pathobiology, University of Toronto, Toronto, ON M5S 1A8, Canada

**Keywords:** calorie restriction, dietary intervention, intermittent fasting, metabolic health, time-restricted feeding, post-dietary, weight regain

## Abstract

Dieting regimens such as calorie restriction (CR) are among the most commonly practiced interventions for weight management and metabolic abnormalities. Due to its independence from pharmacological agents and considerable flexibility in regimens, many individuals turn to dieting as a form of mitigation and maintenance of metabolic health. While metabolic benefits of CR have been widely studied, weight loss maintenance and metabolic benefits are reported to be lost overtime when the diet regimen has been terminated—referred to as post-dietary effects. Specifically, due to the challenges of long-term adherence and compliance to dieting, post-dietary repercussions such as body weight regain and loss of metabolic benefits pose as major factors in the efficacy of CR. Intermittent fasting (IF) regimens, which are defined by periodic energy restriction, have been deemed as more flexible, compliant, and easily adapted diet interventions that result in many metabolic benefits which resemble conventional CR diets. Many individuals find that IF regimens are easier to adhere to, resulting in fewer post-dietary effects; therefore, IF may be a more effective intervention. Unfortunately, there is a severe gap in current research regarding IF post-dietary effects. We recognize the importance of understanding the sustainability of dieting; as such, we will review the known physiological responses of CR post-dietary effects and its potential mechanisms through synthesizing lessons from both pre-clinical and clinical studies. This review aims to provide insight from a translational medicine perspective to allow for the development of more practical and effective diet interventions. We suggest more flexible and easily practiced dieting regimens such as IF due to its more adaptable and practical nature.

## 1. Introduction

As the rates of obesity and its associated metabolic diseases continue to rise globally, there is a pressing need to develop disease management strategies that are both effective and comprehensive. To combat obesity and its downstream metabolic consequences, many attempt to mitigate these effects through dieting interventions such as calorie restriction (CR) [1]. Calorie restriction (CR) has been the most widely practiced form of diet intervention, where the daily total caloric intake is reduced without malnutrition [2,3]. The benefits of CR on metabolism and cardiovascular health are well appreciated [4,5,6]. Unfortunately, the benefits from dieting are often unsustainable in the long term largely due to the termination of dieting regimens, referred to as post-dietary conditions hereafter [7]. Long-term adherence and compliance to CR is challenging and has been shown to create many post-dietary effects [4]. In the context of this review, we use a comprehensive definition of post-dietary effects that encapsulates body weight regain and loss of metabolic benefits. Weight regain is the quick repossession of body weight that was intentionally lost during a dieting intervention, and metabolic phenotypes include parameters such as insulin resistance, glucose tolerance and energy expenditure. Post-dietary CR studies have reported weight regain and loss of metabolic benefits [4,5,6]. Despite the benefits of CR, long term commitment to this diet remains a challenge for many individuals [8,9]. Several clinical studies have reported high dropout rates or insufficient follow-up data due to difficulties of adherence to a CR regimen [10]. The constant episodic change in body weight, defined as weight cycling, can increase the risks of developing metabolic and cardiovascular diseases [11,12].

Due to CR’s compliance challenges, there has been a shift in interest towards intermittent fasting (IF) regimens, such as time-restricted feeding (TRF) and alternate-day fasting (ADF) [13,14]. IF interventions are defined by periodic cycling of fasting and feeding periods, which is deemed to be a more practical diet because it encompasses the freedom to be performed in the presence or absence of CR [15]. IF studies have indicated that even under an isocaloric state, subjects have improved insulin sensitivity, glucose tolerance, and improved response to oxidative stress [16]. Although these increasingly popular fasting regimens have shown beneficial effects in both pre-clinical and clinical studies [17,18,19,20,21], mechanisms underlying post-dietary weight regain and metabolic derangements are not well understood. CR studies in mouse models have suggested that post-dietary body weight regain and metabolic impairments are associated with mechanisms driven by the gut microbiome, hormonal changes, and adipose tissue biology (summarized in Figure 1). Due to the current lack of post-dietary studies in the field of IF, a burning issue in this field is the lack of understanding regarding the post-dietary effects of IF. Therefore, in this article, we review and synthesize the lessons learned from post-dietary CR studies in rodents and humans to provide insight into the post-dietary mechanisms in order to predict the physiological response in IF post-dietary studies.

## 2. Post-Dietary Effects in Mouse Models

Prior to discussing post-dietary effects of CR, it is important to understand the known benefits and mechanisms of CR. CR involves a reduction in caloric intake, which is often achieved through lowering total caloric intake to an amount below total energy expenditure with a goal of body weight reduction and metabolic improvement [22]. CR studies in pre-clinical models have shown many promising effects on metabolic health, obesity, and reduced risk factors of age-associated diseases such as cancer and atherosclerosis [23]. Moreover, studies have also revealed complex mechanisms that involve multi-system crosstalk between the gut microbiome, endocrine system, and adipose tissue, which contributes to the health and metabolic benefits of CR [24,25]. It has been reported that the gut microbiome is a critical factor which directly affects energy harvest from diet and regulates fat storage in the host [24]. Depletion of gut microbiota rendered mice resistant to CR-induced loss of body weight, along with an increase in fat mass, a reduction in lean mass, and a prominent decline in metabolic rate [2], emphasizing the importance of the gut microbiome in CR-mediated metabolic benefits. Furthermore, a previous study demonstrated CR-induced metabolic improvements through adipose tissue alterations, such as a reduction in adipocyte size, increased browning of white fat, increased expression of genes associated with thermogenic functions of brown fat (e.g., uncoupling protein 1, a key thermogenic gene, UCP1) [25]. Browning of adipose tissue is recognized as a common phenotype in conditions of negative energy balance and metabolic improvement.

Compounding what is already known about CR, there has been overwhelming attention and curiosity surrounding post-dietary effects. The underlying mechanisms of post-dietary effects remain unclear, despite postulations that suggest weight cycling is associated with increased disease risk and may pose a major metabolic health concern. Therefore, understanding this phenomenon and its possible drivers and mechanisms are important to provide insight concerning dieting and disease risk. Much of the current pre-clinical post-dietary studies subject mice to CR with a 30% reduction in normal caloric intake or diet-switch regimens that alternate between a high-fat diet (HFD) and a low-fat diet (LFD), commonly referred to as the “yo-yo diet” [26,27]. This review discusses these two dietary regimens because they are currently the most commonly studied regimens in the field of weight cycling and post-dietary research. Based on the current literature in this field, the duration of mouse studies investigating the effects of post-dietary restrictions ranges between 4–6 weeks following termination of dietary restrictions. A study subjected male C57BL6/N mice to a yo-yo diet for a 16-week experimental period to determine whether they would display an altered metabolic status compared to mice on a non-cycling diet regimen [26]. The yo-yo diet in this study was defined as a cycle of eight weeks on an HFD, followed by four weeks on normal chow (NC), then followed by another four weeks on an HFD. Mice subjected to the non-cycling diet regimen were fed NC for 12 weeks, followed by an HFD in the last four weeks of the experiment. The results from this study revealed that the yo-yo dieting mice had more pronounced metabolic derangements in the liver, increased blood glucose levels, and overall metabolic dysfunction compared to the mice with a single exposure of the HFD. In another study, mice subjected to the same 16-week yo-yo diet as described above [27], displayed greater weight regain following initial weight loss which resulted in increased fat mass and adverse metabolic effects. Intriguingly, mice fed a yo-yo diet had larger white adipose tissue depots and greater adipocyte size, elevated fasting blood glucose, and exhibited liver steatosis, compared to isocaloric control mice. In another study, mice were induced to weight gain with an HFD, then subjected to weight loss with a low-fat diet (LFD) [28]. The results revealed that the rate of body weight regain when re-exposed to the HFD was faster than the initial weight gain. Furthermore, the weight regain was positively correlated with the amount of weight lost during LFD feeding. Taken together, this implies that the mice experiencing the greatest diet-induced weight loss may actually be the most vulnerable to rapid weight regain when re-exposed to an *ad libitum* (free access to food) HFD.

Variability in genetic backgrounds are also investigated in CR studies. Although they are not the focus of this paper, they play an undeniable role in determining the effectiveness of CR-mediated responses in mouse models. CR is shown to decrease IGF-1 (insulin-like growth factor 1), which increases apoptosis and inhibits cell proliferation, thereby delaying tumour progression and expanding longevity. Interestingly, sexual dimorphism is reported in IGF-1 expression [29]. When fed ad libitum, elevated IGF-1 expression levels were reported in females compared to males, which continued to remain true under CR conditions [30]. Furthermore, CR improved skeletal muscle function and increased satellite cells in response to aging. However, these changes were not consistent between different mouse strains—while C57Bl/6 increased longevity under CR, DBA/2 showed no significance [31]. These finding emphasize the importance of different genetic backgrounds in determining the effects of CR. It is certainly possible that genetics are involved in regulating post-dietary effects; however, due to the current scarcity in relevant literature, the impact of genetic components was not further discussed in this review.

Considering the CR-mediated alterations of the gut microbiome, hormone levels, and adipose tissue remodeling in contributing to metabolic benefits, it would be reasonable to speculate that post-dietary effects of CR may also be attributed to a reversal of these alterations to conditions prior to CR dieting. The notion of post-dietary effects is deemed as the consequences following termination of dieting regimens. In this review, we explore possible drivers such as behavioural and organ system alterations, which contribute to the overall metabolic effects of post-dietary restriction. Taken together, these observations point to the importance of understanding the pathways involved in post-dietary effects in order to develop effective and sustainable diet intervention programs.

## 3. Possible Drivers of Post-Dietary Effects in Mouse Models

Although diet interventions often result in successful short-term weight loss and metabolic benefits, these benefits are rarely maintained over time. As such, long-term maintenance of weight loss and the sustainability of metabolic benefits remain important challenges for the treatment of obesity and metabolic disorders. Evidence from mouse models indicates that diet-induced weight loss is associated with enduring behavioural changes such as decreases in activity and intrinsic motivation towards food, thereby contributing to chronic obesity, metabolic dysfunction and relentless post-dietary weight regain [28]. In light of these findings, combined with its clinical relevance, research aimed to identify and better understand the mechanisms driving post-dietary effects are needed urgently. In particular, the gut microbiome, hormone secretion, and adipose tissue remodelling have been reported to be responsible for driving CR-mediated metabolic alterations [2,4,32]. Based on the intestine’s central role in nutrient absorption and digestion, intestinal homeostasis is crucial in the maintenance of systemic metabolic health through the regulation of the gut microbiome [33,34]. The gut microbiome has proven to play a central role due to its diverse microbial populations, regulation of metabolic hormone secretion, energy harvesting, and adipose tissue biology [2,4,35]. Collectively, the gastrointestinal tract, microbiome, and adipose tissue have a broad-spectrum impact on whole-body metabolism. It is speculated that these are key drivers associated with post-dietary metabolic health parameters such as weight management, insulin resistance, glucose tolerance and energy expenditure.

### 3.1. Microbiome Alterations

The gut microbiome is one of the key organs involved in energy harvesting and regulation of host metabolism [35]. In a study where researchers compared germ-free mice with mice transplanted with normal gut microbiota, the results revealed that the microbiome-transplanted mice exhibited a significant increase in body fat mass by 60% [24]. Another study reported that the microbiome of *ob*/*ob* mice could harvest more energy compared to their lean counterparts [5]. These findings emphasize the gut microbiome as an important factor in the regulation of energy harvesting and metabolism.

In a study to investigate the functional role of the gut microbiota in a CR model, antibiotic-induced microbiota-depleted mice were placed on either a normal ad libitum diet or a 30%-reduced CR diet [2]. Depletion of the microbiota rendered mice resistant to CR-induced body weight reductions and resulted in a gain of fat mass, loss of lean mass, and perturbed metabolic rate. Furthermore, based on the Shannon and Sobs indexes, the gut microbiota of a CR diet had a more diversified and well-balanced microbiome ecosystem [2]. Low gut bacterial richness and diversity is common to a plethora of chronic diseases in both mice and humans [36,37]. Beneficial bacterial phyla such as *Bacteroidetes* and *Actinobacteria* were increased, while harmful *Firmicutes* and *Verrucomicrobia* were slightly decreased. Widely accepted probiotics associated with health-promoting and immunomodulatory properties such as *Lactobacillus* and *Bifidobacterium* [38,39] had increased proportions in CR mice [2]. *Helicobacter*, a pathogenic bacteria which predominantly resides in the upper gastrointestinal tract [40], was significantly reduced by CR [2]. Fecal microbiota transplantation (FMT) in diet-induced obese mice was used to further investigate whether gut microbiota changes during CR are causally associated with metabolic improvements in mice [2]. These results revealed that the mice receiving FMT from CR treated mice exhibited a reduced gain of body weight, decreased fat mass, and significant reductions in their fasting blood glucose and blood leptin level, which were largely consistent with results from CR mice [40]. Together, this highlights the microbiome’s pivotal role in mediating diet-induced improvements in metabolism, infectious control, and health promotion.

Considering the highly sensitive nature and persistent effects of alterations in the gut microbiome, its contribution to the long-term outcomes of dieting is compelling [41]. In a study to investigate post-dietary weight regain, mice were subjected to a recurrent obesity model where mice were exposed to cycles of HFD, interleaved with normal diet consumption [4]. Mice fed with cycles of HFD experienced weight gain and developed metabolic syndrome during the primary exposure to an HFD. Interestingly, recuperation of the metabolic syndromes and weight reduction was observed during the period of normal diet consumption but was followed by a re-emergence of weight gain and associated metabolic disturbances from subsequent cycles of HFD-induced obesity. Recurrent obesity was characterized by enhanced glucose intolerance, elevated serum levels of leptin and low-density lipoprotein (LDL), an increase in total body fat composition, and decreased energy expenditure [4]. These results demonstrated exacerbated metabolic derangements, suggesting that the previous dieting cycles progressively promoted the susceptibility of accelerated weight regain and is closely linked to various metabolic complications. Notably, an intermediate configuration of the microbiota between normal and dysbiotic states were observed during HFD and normal diet consumption periods, whereby the bacterial alpha diversity was reduced significantly. The bacterial population was altered during the induced obesity state and did not recover even after normal body weight and metabolic homeostasis were achieved. In line with this, other studies have also identified microbiome dysbiosis as a strong contributor to the pathogenesis of obesity and metabolic disorders [5,42,43]. Collectively, this highlights the microbiome contribution to post-dietary weight regain and associated metabolic complications.

The gut microbiome has an active and integral role in regulating metabolic hormones secreted by organs such as the gastrointestinal tract and adipose tissue. It has been shown that depletion of the gut microbiome greatly affects hormone secretion, thereby leading to detrimental effects to metabolic health [2,44,45]. A study observed hyperphagia in antibiotic-treated mice and revealed the effects of microbiota-mediated alterations on hormone secretion that regulate whole-body metabolisms, such as gastric inhibitory polypeptide (GIP) and PYY [2]. PYY is a hormone produced in the small intestine which functions to limit food intake through appetite suppression [46]. The endocrine functionality of the gut microbiome plays a critical role in CR-mediated benefits, such as a reduction in body weight, maintenance of high basal metabolic rate, and reduction in blood glucose levels. Specifically, the FMT experiment in this hyperphagia study demonstrated that gut microbiomes from CR mice can alleviate obesity in diet-induced obese mice [2]. Antibiotic treatment largely abolished the metabolism-regulator endocrine function of the gut microbiome and led to abolishment of CR-mediated health benefits. Although dietary induced cross-talk between the gut microbiota and endocrine system is known, much of this cross-talk relationship is still unknown under post-dietary conditions. Further investigation of the post-dietary effects of CR-mediated hormonal changes are required to better understand its implications following termination of a dietary intervention.

Excess caloric intake is commonly regarded as one of the root causes of obesity, although a distinction in gut microbial populations has been deemed a prominent factor affecting energy, metabolic and endocrine homeostasis. In particular, significant differences in the microbial composition between obese and lean mice have been reported and well-studied [36]. It has been suggested in both pre-clinical and clinical studies that predisposition to obesity comprises gut microbial populations that promote more extraction and/or storage of energy from a given diet, thereby leading to greater weight gain compared to the microbial populations of lean subjects [36,47]. A study revealed a significant 50% reduction in *Bacteroidetes* and a significantly greater proportion of *Firmicutes* in obese mice compared to their lean counterparts [36]. Altogether, these results provide a supportive rationale that the gut microbial composition has fate-determining effects on post-dietary weight management and overall metabolic health.

### 3.2. Adipose Tissue Remodelling

Mechanistic changes in adipocytes are proposed to be a prominent factor driving post-dietary weight regain [48]. White adipose tissue is known to be the reservoir for energy storage, but they also carry secretory and endocrine functions [49]. Obesity induces significant changes in the structure and function of adipose tissue. Lean mice are associated with M2-like macrophages, and other anti-inflammatory adipokines (e.g., adiponectin) are present. In comparison, obese mice are accompanied by M1-like pro-inflammatory macrophages along with other pro-inflammatory cytokines such as resistin, leptin, TNF, and IL-6 [50].

In a weight cycling study, mice were subjected to a diet-switch model where mice were fed HFDs, followed by normal chow, then rechallenged again with HFDs to induce weight regain [32]. Results from this study revealed that weight gain in formerly obese mice might accelerate the development of liver steatosis due to epidydimal white adipose tissue (eWAT) dysfunction. The failure of adipose tissue remodelling to permit healthy adipose expansion continued up to six months after weight loss, and further exacerbated liver steatosis and liver dysfunction. The observed insulin resistance marked by a two-fold increase in fasting glucose levels, combined with previous evidence of liver steatosis and liver insulin signal impairment in weight-cycling models underscores the role of adipose tissue dysfunction in amplifying the development of metabolic abnormalities. Despite the quantitative increase in preadipocytes, the increased pro-fibrotic *Col1*+ preadipocytes suggest that they are taking on a myofibroblast phenotype with impaired adipogenic differentiation capacity, consequently leading to limited capacity for eWAT expansion and dysfunction [32]. Mature adipocytes are crucial in regulating energy storage and balance. However, various critical mature adipocyte-specific genes such as *Pref1*, *Col1a1*, *Col3a1*, *Acta2*, and *Fnl* exhibited significantly reduced expression in formerly obese eWAT, which remained at low levels following the HFD rechallenge. This suggests that mice subjected to HFD rechallenge had significantly impaired lipid storage capacity. Overall, the results demonstrate that adipose tissue of obese and formerly obese mice contribute to impaired adipogenesis and the failure to induce new properly functioning adipocyte formation when subjected to HFD rechallenge. This is consistent with previous studies, which demonstrated that prolonged recovery from HFD did not recover the adipose tissue environment, insulin sensitivity, or lipid storing functionality that was disrupted during previous periods of HFD or obesity [51,52]. Together, the physiological differences associated with weight regain and metabolic impairment in formerly obese mice highlight that obesity may result in long-term impairment of adipose tissue function. Further work is needed to investigate the specific roles of adipose tissue in a post-dietary context.

The hypothalamus is the central site where many neuropeptides and hormones act to elicit appetite regulation [53]. However, the hypothalamus also has peripheral signals responsible for achieving energy homeostasis through integrating information such as short-term food intake and long-term energy balance signals [53,54]. These processes occur through a feedback loop between the brain and periphery tissues such as the gastrointestinal tract, pancreas, liver, muscle, and adipose tissue. The commonly examined hormones include peptide YY (PYY), glucagon-like peptide-1 (GLP-1), and cholecystokinin (CCK) from the gastrointestinal tract, insulin from the pancreas, and leptin from adipocytes [2,46,55,56,57]. Gastric inhibitory polypeptide (GIP) is an obesity-promoting factor that acts on adipocytes [55,56]. The plasma levels of PYY and GIP were significantly higher in antibiotic-induced microbiota-depleted CR mice compared to CR mice with uncompromised gut microbiota. Leptin is a hormone released by adipose tissues which send signals to the hypothalamus and regulate energy balance [57]. Energy balance is regulated through the suppression of hunger, and deficiencies in leptin are often associated with obesity in both mice and humans [58,59]. This study revealed a reduction in plasma levels of leptin in CR mice; this effect was further enhanced when mice were subjected to antibiotic treatment, suggesting the importance of the microbiome in leptin regulation [2]. Insulin is also known as an acute appetite suppressant [60], thereby it helps the body properly store glucose through the metabolic regulation of proteins, lipids, and carbohydrates. Plasma levels of insulin were significantly decreased following antibiotic treatment, which was compounded with elevated fasting blood glucose levels upon antibiotic administration [2]. It is important to acknowledge the effects of CR-induced hormonal changes on the multi-organ communication system because of its inevitable downstream effects on whole body metabolism and homeostasis.

Evidently, body weight maintenance and metabolic homeostasis are intricately regulated by processes such as multi-organ cross-talk, and environmental and behavioural factors. The hypothalamus has a central role in maintaining homeostasis by integrating signals pertaining to energy balance, food intake, and body weight in both pre-clinical and clinical models [61,62,63]. Thus, a more thorough and comprehensive physiological understanding of post-dietary effects and the sustainability of metabolic benefits are critical for the development of more effective weight management strategies.

## 4. Post-Dietary Effects in Clinical Settings

Dietary interventions are commonly being practiced as non-pharmaceutical treatments for tackling obesity and metabolic diseases; therefore, the need for a more thorough understanding of the benefits and drawbacks of dieting is more imperative than ever. The conventional interventions containing energy-deficit diets such as CR emphasize the critical role of caloric intake in the management of body weight, fat mass, and metabolic abnormalities. Evidence has indicated that CR diets modify cardiometabolic conditions, thereby greatly reducing various risk factors of age-related and cardiovascular diseases, cancers, and metabolic syndromes [3,64]. However, compliance with conventional CR interventions have been proven to be a major challenge in clinical settings. A remarkable proportion of individuals worldwide attempt to lose weight; unfortunately, many are confronted with poor long-term results [65,66]. Importantly, the challenges in maintaining body weight reduction are believed to drive individuals in making repeated attempts at managing their weight through cycles of yo-yo dieting, characterized as periodic cycles of on and off dieting [7]. These repeated attempts at weight loss often leads to weight cycling, a perpetual cycle of weight loss followed by weight gain. The high likelihood of weight regain and relapse of metabolic benefits emphasize the pressing need to better understand the underlying mechanisms of post-dietary effects in clinical settings.

Despite this dire need, clinical studies examining the prevalence of weight cycling are limited and the outcomes are relatively inconsistent. While some studies have suggested that the regaining of body weight following diet-induced weight loss is associated with an increased risk for stroke, diabetes, coronary heart disease, and mortality [7,67,68], other studies have reported no detrimental effects from weight cycling in obese individuals [69]. Two meta-analyses examined several weight cycling studies and reported no major adverse effects [41,69]. The definition and duration of weight cycling are widely varied across clinical studies, and the targeted populations are also considerably inconsistent. Furthermore, due to the limited clinical studies investigating post-dietary effects, the methodology, duration, and study designs are rather inconsistent. Based on the current literature in the field, the duration of follow up after termination of dietary restriction ranges from four weeks to six months.

Humans exhibit complex and distinctive genomes, and as such, CR may have diverse effects on individuals due to genomic variations. It was demonstrated that single nucleotide polymorphism of the peroxisome proliferator-activated receptor γ (PPARG) gene was responsible for approximately 7% of total reduced body weight variance during a short-term CR regimen (14 weeks) [70]. Furthermore, the *Ala55Val* genetic polymorphism in uncoupling protein-2 (UCP-2) was associated with abdominal subcutaneous fat, where *ValVal* types showed significantly less reduction under CR compared to *AlaAla* or *AlaVal* types [71]. These findings suggest that genomic differences among individuals may affect the outcome of CR. Thus, the post-dietary effects may also have genetic components involved to affect the severity and duration, which requires further investigation.

As such, the outcomes of current weight cycling studies are largely impacted by the multiple confounding variables. It is rather difficult to isolate the impact of the post-dietary effects. Therefore, a consensual methodology and a more thorough understanding of post-dietary effects are needed to provide insightful, practical, and effective treatments to address obesity worldwide.

### 4.1. Human Microbiome Alterations

The post-dietary effects of weight cycling in clinical settings remain inconclusive and unclear. The underlying mechanisms need to be further investigated to develop effective, comprehensive, and evidence-based treatment strategies for obesity and metabolic diseases. Due to the resemblance in genomics between mice and humans, the gut microbiome is heavily studied in clinical studies. Similar to rodent studies [4], the microbiome is considered a key regulating factor of dietary responses and fat storage; therefore, it may also be largely responsible for post-dietary effects in humans.

Gut microbiota in humans have been reported to be dynamically altered by both CR and nutritional components [72,73]. Thus, it is not surprising that the gut flora between a meat and non-meat consumer is also vastly different [74]. *Firmicutes* and *Bacteroidetes* are the two dominant beneficial bacterial strains residing in the gut microbiome [75]. Interestingly, the gut microbiota composition of 12 obese individuals were monitored through sequencing 16S ribosomal RNA genes over a one-year period. During this period, participants were subjected to a CR diet, either with a fat-restricted or carbohydrate-restricted low-caloric intake. Before CR, individuals had increased levels of *Firmicutes* compared to *Bacteroidetes*. After dieting, a dramatic shift in ratio was reported; reduced levels of *Firmicutes* and elevated levels of *Bacteroidetes*. The ratio of *Firmicutes*:*Bacteroidetes* in obese individuals shifted towards becoming more similar to lean controls following a CR diet, irrespective of macronutrient content. On the other hand, other clinical studies have demonstrated the dynamic and rapidly adaptive nature of the gut microenvironment [76,77]. In a short 10-day fasting study, fecal samples of fifteen healthy men were examined both before and after fasting [76]. Decreased abundance of *Lachnospiraceae* and *Ruminococcaceae,* and increases in *Bacteroidetes* and *Proteobacteria* were reported, which highlights that certain residing bacterial strains are more sensitive to dietary interventions, and more specifically that the alterations in microbial populations are quickly adaptive. Thus, post-dietary weight regain loss of metabolic benefits could be the result of the active remodeling of the microbiome, further favoring fat storage. However, although microbiota-mediated post-dietary weight regain have been investigated in rodent studies [4], whether this is also true in humans still requires further investigation.

### 4.2. Hormonal and Behavioural Alterations

Body weight and metabolic homeostasis are mediated by the complex neuro-hormonal system, which coordinates peripheral anorexigenic or orexigenic neuropeptides and hormones [78]. The arcuate nucleus, located within the hypothalamus, is the main site where the peripheral signaling and feedback are integrated [79]. An intricate network of antagonistic neurons and peptides work together to elicit various feeding behaviours. In clinical models, some specific examples are: agouti-related peptide (AgRP)/neuropeptide Y (NPY) neurons, which promote feeding behaviours; and pro-opiomelanocortin (POMC) neurons, which suppress feeding [79,80]. Furthermore, many neuropeptides and hormones mentioned in the rodent section of this review also play a central role in humans.

Leptin secreted from adipose tissue predominantly act on POMC neurons in the hypothalamus to induce satiety. In a study where the obese participants underwent a weight-loss program, overall weight loss resulted in a reduction in leptin signaling, which led to reduced or delayed feeding inhibition in the study subjects [81]. Following replacement doses of leptin through subcutaneous injections, subjects experienced increased satiation. Taken together, leptin depletion following weight loss exhibited a blunted expression on feeding inhibition and satiety. Furthermore, gut-secreted hormones, including ghrelin, PYY, CCK, GIP, and GLP-1, have also been examined in human studies [53,61,63,78]. Ghrelin, originating from the stomach, is an appetite-stimulating hormone; levels of ghrelin decrease following feeding [53]. Intestine-secreted CCK (cholecystokinin) delays gastric emptying and stimulates the breakdown of fats and proteins [53]. Alterations in these hormone levels have been shown to promote appetite and energy storage during post-dietary weight regain [53]. Typically, it is reported that following a diet-induced weight-loss program, elevated levels of ghrelin and GIP result in increased feelings of hunger [61,63,78]. Leptin, PYY, CCK, and GLP-1 are reduced, which may lead to faster gastric emptying and reduced satiety. In a study which reported CR-induced reductions in leptin, PYY, CCK, and GLP-1 [63], it was reported that these phenotypes were still maintained one year following CR intervention, thereby highlighting the potential long-term adaptations due to CR-induced weight loss. However, while these studies demonstrate that the relevant hormones could be predictors of weight regain and metabolic consequences, there are other studies showing no significant association. This emphasizes the importance of future research aimed at delineating the role of hormones in regulating post-dietary effects.

Humans exhibit a much more complex system, whereby extrinsic factors such as the external environment and lifestyle directly affect behaviours, which leads to fluctuations in body weight and metabolic homeostasis. Due to food being conveniently accessible with our modern-day societies, hunger is certainly not the only factor regulating energy intake. Thus, hedonic reward pathways are commonly investigated in clinical settings as it involves cognitive, reward, and emotional factors [80]. Palatability, visualization, and smell drive cravings and lead to the unconscious consumption of high-calorie foods containing high fat and sugar content [53]. The dopamine-mediated reward system and opioid-mediated reward system are two main pathways which associate obesity with food addiction [82]. Furthermore, different environmental factors drive food consumption choices and eating patterns that affect daily choices [61]; therefore, it is rather difficult to predict and delineate the unique impacts of such intertwined factors in clinical setting. Unlike pre-clinical studies, clinical studies have many more extraneous variables that must be taken into consideration. As such, inconsistencies are commonly reported among research groups. Therefore, novel reviews and comprehensive meta-analyses are needed to further strengthen the currently known phenotypes and to discover new post-dietary mechanisms in order to provide more meaningful insight on guiding the future of bedside research.

### 4.3. Adipose Remodeling in Humans

Similar to reports in rodent models, human adipose tissue alterations are proposed to be involved in the mechanisms driving post-dietary weight regain, because it plays a major role in fat storage [48,83]. Adipose tissues in obese individuals have been infiltrated with M1 macrophages, thereby exhibiting inflammatory responses [50]. This study collected subcutaneous abdominal fat from obese subjects for immunochemistry analyses, which revealed that inflammatory adipose tissues possessed crown-like structures and were infiltrated with M1 macrophages. Furthermore, hyperinsulinemia, insulin resistance, and vascular impairments have also been reported, which emphasizes adipose inflammation and its link to increased cardiometabolic risks [84].

Dynamic adipose tissue remodelling has been reported as a result of calorie restriction and overfeeding. A study investigating the initial period of weight gain induced by overfeeding has reported that individuals had an upregulation in genes associated with lipid metabolism and storage, angiogenesis, and extracellular matrix remodelling [85]. Several studies [86,87,88,89], have reported and proposed plausible mechanisms mediated by adipose changes associated with weight regain [48,83,90]. While the adipocyte stress models differ slightly among studies, the central idea remains similar, suggesting that although the adipocytes undergo reductions in size due to caloric restriction, the adipocytes and the extracellular matrix (ECM) must be remodeled in order to accommodate the immediate change accordingly [83,87]. However, ECM remodelling is arduous under low-calorie intake because less excess energy is available for utilization. Due to the adipocyte shrinkage and the inability of ECM to remodel, the mechanical stress is built up at the focal adhesions between the adipocytes and ECM, and is only relieved if adipocytes are able to revert to their original cell volume prior to dieting conditions, which drives weight regain in the host [48,83,87,88]. It has also been reported in several clinical trials that the cellular stress induced by weight loss in subcutaneous adipose tissue is related to weight regain [88,91,92]. A study recruited 61 overweight or obese individuals and randomly assigned them to one of two dietary interventions with differing durations and caloric intake, either a very-low-calorie diet (500 kcal per day) or a low-calorie diet (1250 kcal per day) [91]. After this CR diet period, a follow-up was performed throughout a nine-month period. It was reported that the very-low-calorie diet group had a stronger correlation with an increase in cellular stress proteins compared to the low-calorie diet group; however, both groups revealed changes in the expressions of adipocyte cellular stress-related genes associated with risks for weight regain [91]. Interestingly, similar trends were observed in healthy individuals under post-dietary conditions [92]; more cellular stress protein accumulation such as β-actin, HSP60, and HSP70 are reported following weight regain. Together, this highlights that under post-dietary conditions, independent of baseline metabolic health status, increased levels of cellular stress proteins were reported.

Many other diet-induced mechanisms of adipose tissue remodelling are being investigated in post-dietary studies, such as its metabolic capacity, endocrine functions, and inflammatory responses [48,87]. For example, lipolysis is a process involved in mobilizing stored energy in adipose tissue and has been a mechanism used to predict the onset of weight gain [48,93,94,95]. A study comparing phenotypes of a weight-stable and weight-gain group reported that individuals who experienced weight-gain over a long-term period of 10 years exhibited lowered expression of several lipolysis-regulating genes. The weight-gain group demonstrated inefficient lipolysis along with the progressive development of impaired glucose metabolism [93]. Thus, inefficient lipolysis processes may be an indicator of post-dietary effects. As a result, while the evidence of adipose-associated post-dietary effects are observed, more thorough understanding of adipose tissue alterations in humans are required to fully delineate the diverse impacts of post-dietary induced remodelling phenotypes and to investigate its downstream metabolic effects.

## 5. Discussion and Future Directions

Inconsistencies and gaps in the current literature are rooted in many limitations and challenges associated with the current research on post-dietary weight management and metabolic effects. Some of these barriers include inconsistent definitions and definitive terminology, varying targeted populations, and lack of unified methodologies, thus resulting in major sources of bias between pre-clinical and clinical studies. In the current literature, there are many different key terms used to characterize post-dietary weight regain and metabolic phenotypes [4,26,28,61,90]. Studies also often utilize slightly different parameters for these measures. Thus, this lack of consistency used to define and characterize the post-dietary weight regain phenomenon and loss of metabolic benefits poses significant limitations and confounders to post-dietary research. These sources of inconsistency in both pre-clinical models and human trials make it extremely difficult to establish a strong independent correlation between post-dietary effects and its related risk factors. Compared to pre-clinical models, where post-dietary studies can be followed up and monitored following the termination of dieting in a stringent and systematic manner, clinical post-dietary studies often have inconsistent results due to self-reported data [65], high drop-out rates [66], small sample sizes [34], and short follow-up periods [76]. More rigorous and long-term post-dietary studies are required to observe clinically accurate weight management and metabolic phenotypes. More importantly, adherence and proper compliance to diet regimens are difficult to monitor in clinical studies due to the nature of human tendencies. Dieting regimens may not be a practice in which individuals commit indefinitely, thus highlighting the importance in understanding how switching on and off a dieting regimen could affect metabolism and other health parameters to follow. Therefore, investigating dietary practices that allow for greater flexibility and customizability would shed light on the efficacy of diet-focused interventions for the treatment of obesity and metabolic diseases.

IF regimens and their abundance of variations have revealed many benefits that resemble conventional CR regimens. Early time-restricted feeding (eTRF) has been commonly used as a form of IF in human studies. Sutton et al.’s study [16] demonstrated that a short five-week isocaloric eTRF was sufficient to elicit benefits to cardiometabolic health in prediabetic men. IF was able to improve insulin sensitivity, lower blood pressure, and reduce oxidative stress. Participants did not report a significant reduction in their body mass; as such, the cardiometabolic benefits were deemed independent to weight loss. In another study, eTRF was conducted over a shorter duration of four days and revealed reduced average 24 h glucose levels and increased fat oxidation [96,97]. Alternate day fasting (ADF) is another variation of IF, characterized by alternate fasting and feeding days. In a study comparing ADF and CR regimens, the results of an eight-week diet intervention followed by a 24-week post-dietary follow-up were examined to assess the risk of weight regain and loss of metabolic improvements [98]. Both ADF and CR groups revealed significantly decreased levels of total cholesterol, HDL, and LDL, as well as fasting glucose. Interestingly, throughout the unsupervised 24 weeks of post-dietary follow-up, the ADF group lost fat mass and gained lean mass, while the CR group gained both fat and lean mass [98].

Although IF has been shown to be a promising diet intervention in pre-clinical and clinical models, further research is needed to delineate the post-dietary effects of IF. While IF regimens are flexible, more easily adaptable to suit various lifestyles and often include isocaloric consumption, CR regimens mainly focus on chronic and consistent caloric restriction which likely drives adapted responses in metabolic organs and systems such as adipose tissue, gut microbiome, and hormone levels. Due to the adaptation of chronically reduced caloric intake, an increase in energy absorption and storage often leads to downstream metabolic impairments. In contrast, due to the ability to maintain isocaloric consumption periods in IF regimens, the efficiency of energy utilization can be sustained. Under IF conditions, metabolic organs and systems are able to be well adapted to consistent levels of caloric intake. Therefore, due to the fundamental difference between IF and CR regimens, the outcome of post-dietary effects of IF may also be different from CR. While we can synthesize, extrapolate, and apply the mechanisms known from CR post-dietary research, more work is required to investigate the metabolic effects of IF post-dieting among both pre-clinical and clinical models.

## Figures and Tables

**Figure 1 metabolites-11-00062-f001:**
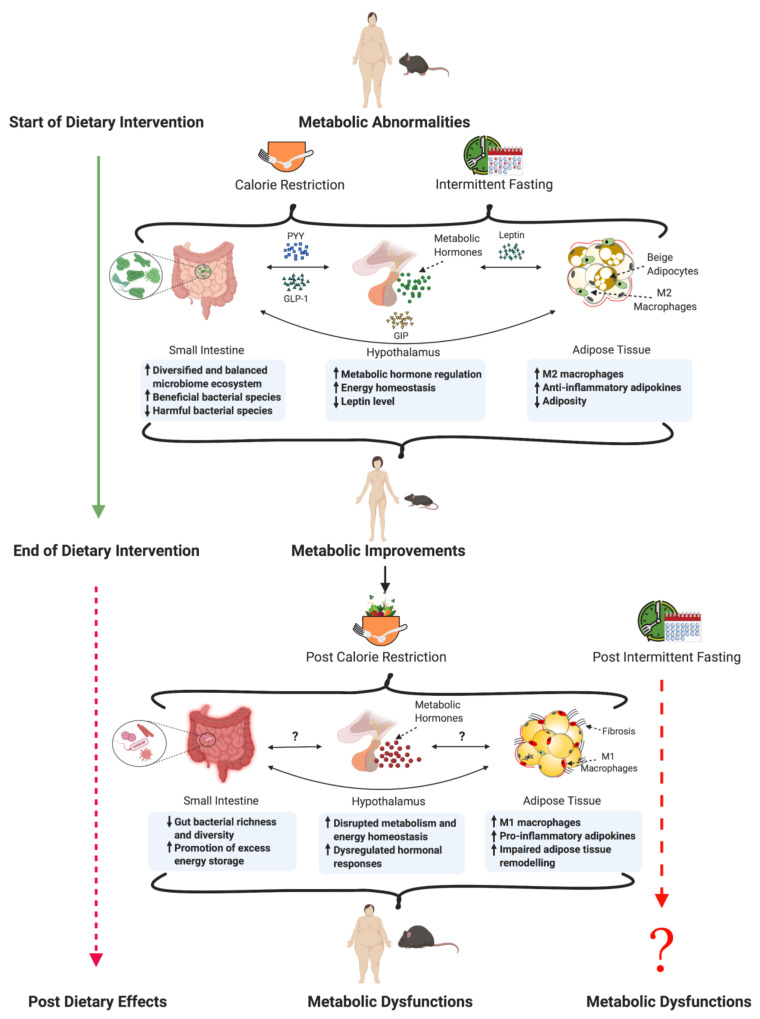
Schematic diagram summarizing the metabolic post-diet effects of CR and IF regimens in mouse and human studies. (Created with BioRender.com).

## Data Availability

Not applicable.

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
