# Peer review of "Physiological Responses of Post-Dietary Effects: Lessons from Pre-Clinical and Clinical Studies"

_metabolites, 2021, doi:10.3390/metabo11020062_

Round 1

Reviewer 1 Report

The manuscript entitled “Physiological responses of post-diet effects: lessons from pre-clinical and clinical studies” reviews recent findings on the physiology after the termination of dieting regimens. The authors raise important concerns regarding post-diet repercussions found in clinical and pre-clinical studies. Discussions in the review article provide mechanistic explanations for several clinically significant phenomena. Specifically, the microbiome-mediated hormonal alterations and the adipose tissue remodeling are identified as potential drivers of the post-diet effects. Overall, this review provides up-to-date knowledge from a translational perspective, which is of fundamental interest in dietary research and will help understand metabolic diseases and develop therapeutic diets.

I have only a couple of suggestions regarding the definition of “post-diet condition”. Given that the concept of post-diet effects is relatively new and undefined to the field, the authors may want to define the “post-diet condition” in a broader context, in addition to Calorie Restriction (CR), and with regard to the following operational details. First, how long after termination of dietary restriction is considered as the post-diet condition in this review? This clarification is critical as, for example, fasting interventions often include cycles of fasting and refeeding. The effects of refeeding (hours to days after fasting) and post-CR physiology (months to years after long-term CR) may not be comparable and should be characterized separately.  Second, is the altered post-diet eating behavior considered as the cause or the consequence of the post-diet effects?  The current manuscript seems to generally interpret “termination of dieting regimens” as “resuming pre-diet eating behavior”. It may be true for most of the preclinical experiments but not the case in many clinical studies.

The schematic diagram summarizing the key concepts and the outstanding question is very helpful. The author may consider either indicating the organs (i.e. intestine, adipose tissue, hypothalamus) in the figure or describing it in the figure legends. It would also be very appreciated if the authors can incorporate the multi-organ cross-talking in the model figure.

Reviewer 2 Report

The manuscript of Yeung et al. summarizes the literature about post interventional effects of diet with a focus of weight regain and metabolic dysfunction. The manuscript is excellently written and highly interesting contribution to the field. However, there are some aspects which have to be addressed.

  1. The manuscript completely ignores genetic factors modifying caloric restriction, weight gain, weight regain/weight cycling, insulin sensitivity and metabolic dysfunction. This has to be addressed in a seperate section and within the discussion.
  2. Section 3.2 does not summarize anything about post-dietary intervention effects, it should be deleted. I understand that the authors are interested in this specific topic, but it does not fit to the overall topic.
  3. The term “post-diet” is misleading. Post-diet can be understood that there is no diet anymore. ut the animals and humans just change their diets after intervention, e.g. from CR to ad libitum diet or LFD to HFD. It would be more precise to call it “post-dietary intervention”. The text and the title of the paper should be changed accordingly.

Round 2

Reviewer 2 Report

The authors adressed all points raised and the manuscrip has further improved.